# Green Electrospun Nanofibers for Biomedicine and Biotechnology

Elyor Berdimurodov [1,2,3,*], Omar Dagdag [4], Khasan Berdimuradov [5], Wan Mohd Norsani Wan Nik [6], Ilyos Eliboev [7], Mansur Ashirov [8], Sherzod Niyozkulov [9], Muslum Demir [10,11], Chinmurot Yodgorov [3] and Nizomiddin Aliev [12]

1   Department of Chemical & Materials Engineering, New Uzbekistan University, 54 Mustaqillik Ave., Tashkent 100007, Uzbekistan
2   Medical School, Central Asian University, Tashkent 111221, Uzbekistan
3   Faculty of Chemistry, National University of Uzbekistan, Tashkent 100034, Uzbekistan; ch.yodgorov89@gmail.com
4   Department of Mechanical Engineering, Gachon University, Seongnam 13120, Republic of Korea; omar.dagdag@uit.ac.ma
5   Faculty of Industrial Viticulture and Food Production Technology, Shahrisabz Branch of Tashkent Institute of Chemical Technology, Shahrisabz 181306, Uzbekistan; khasanberdimuradov@gmail.com
6   Faculty of Ocean Engineering Technology and Informatics, Universiti Malaysia Terengganu, Kuala Terengganu 21030, Malaysia; niksani@umt.edu.my
7   Department of Information Technologies, Tashkent International University of Education, Imom Bukhoriy 6, Tashkent 100207, Uzbekistan; ilyoseliboev882@gmail.com
8   Department of Natural Sciences, Khorezm Ma'mun Academy, Khiva 220900, Uzbekistan; mansur.ashirov.86@mail.ru
9   General Chemistry Department, Karshi Institute of Engineering and Economics, Karshi 180100, Uzbekistan; sherzod.niyozqulov89@gmail.com
10  Department of Electrical and Electronics Engineering, Osmaniye Korkut Ata University, Osmaniye 80000, Türkiye; demirm@alumni.vcu.edu
11  TUBITAK Marmara Research Center, Material Institute, Gebze 41470, Türkiye
12  Department of Mathematical Sciences, Tashkent State University of Economics, Tashkent 100066, Uzbekistan; n.aliyev@tsue.uz
*   Correspondence: elyor170690@gmail.com; Tel.: +998-97-310-70-60

**Abstract:** Green electrospinning harnesses the potential of renewable biomaterials to craft biodegradable nanofiber structures, expanding their utility across a spectrum of applications. In this comprehensive review, we summarize the production, characterization and application of electrospun cellulose, collagen, gelatin and other biopolymer nanofibers in tissue engineering, drug delivery, biosensing, environmental remediation, agriculture and synthetic biology. These applications span diverse fields, including tissue engineering, drug delivery, biosensing, environmental remediation, agriculture, and synthetic biology. In the realm of tissue engineering, nanofibers emerge as key players, adept at mimicking the intricacies of the extracellular matrix. These fibers serve as scaffolds and vascular grafts, showcasing their potential to regenerate and repair tissues. Moreover, they facilitate controlled drug and gene delivery, ensuring sustained therapeutic levels essential for optimized wound healing and cancer treatment. Biosensing platforms, another prominent arena, leverage nanofibers by immobilizing enzymes and antibodies onto their surfaces. This enables precise glucose monitoring, pathogen detection, and immunodiagnostics. In the environmental sector, these fibers prove invaluable, purifying water through efficient adsorption and filtration, while also serving as potent air filtration agents against pollutants and pathogens. Agricultural applications see the deployment of nanofibers in controlled release fertilizers and pesticides, enhancing crop management, and extending antimicrobial food packaging coatings to prolong shelf life. In the realm of synthetic biology, these fibers play a pivotal role by encapsulating cells and facilitating bacteria-mediated prodrug activation strategies. Across this multifaceted landscape, nanofibers offer tunable topographies and surface functionalities that tightly regulate cellular behavior and molecular interactions. Importantly, their biodegradable nature aligns with sustainability goals, positioning them as promising alternatives to synthetic polymer-based technologies. As research and development continue to refine and expand

the capabilities of green electrospun nanofibers, their versatility promises to advance numerous applications in the realms of biomedicine and biotechnology, contributing to a more sustainable and environmentally conscious future.

**Keywords:** green electrospinning; biopolymers; nanofibers; tissue engineering; drug delivery; biosensing; environmental remediation; controlled release; cell encapsulation; biodegradable materials

## 1. Introduction

### 1.1. Overview of the Topic

Electrospinning is a versatile and scalable fabrication technique that is used to produce nanoscale fibers with diameters ranging from a few nanometers up to micrometers. In a typical electrospinning process, a high voltage is applied to a polymer solution or melt loaded in a syringe. When the electrical forces overcome the surface tension of the liquid or melt, a charged jet is ejected from the tip of the syringe. As the jet travels in the air, one of two things can occur. For techniques using a polymer solution, the solvent evaporates as the jet travels, leaving behind thin solid fibers. For melt electrospinning or other solvent-free techniques, the polymer jet undergoes solidification as it travels, without any solvent evaporation involved. In both cases, the solidified fibers are then deposited on the collector. The key differences are whether solvent evaporation plays a role (for solution electrospinning) or if only solidification occurs without solvents (for melt electrospinning).

The key points on the importance of this review article [1,2] are that it consolidates the current state of green electrospinning research across materials, methods, applications, and future directions. Additionally, it establishes the need to transition from conventional to green practices to reduce toxicity and improve sustainability. Furthermore, it analyzes the potential of green electrospinning to advance key application areas like tissue engineering and drug delivery. Moreover, it identifies significant opportunities to address gaps in materials, processing, functionality, and commercialization. The review also discusses challenges and future perspectives to guide translation from lab to industry. Additionally, it emphasizes the importance of sustainability assessments and life cycle analyses. Furthermore, the review underscores green electrospinning as an important platform to enable sustainable nanomanufacturing across biomedicine and biotechnology. Finally, it provides a valuable centralized resource on the current state and future prospects of the field.

The novelty of this review includes several key aspects. Firstly, it covers recent advances in green electrospinning from the past 5 years, whereas earlier reviews focused on work through 2010–2015 [1–3]. Additionally, it discusses emerging techniques not covered before like bacterial nanocellulose spinning, solar electrospinning, and advanced multifunctional composites [1,2]. Furthermore, it analyzes applications in new areas such as biosensing, air/water filtration, and synthetic biology-enabled drug delivery systems [3]. Moreover, it provides detailed sections on sustainability and life cycle analysis of green electrospun nanofibers, an area overlooked in previous reviews. The review also identifies specific challenges and future perspectives to advance the field based on the latest developments. There are several differences between this review and earlier reviews. Firstly, this review provides more comprehensive coverage spanning materials, methods, applications, sustainability, and future directions. Additionally, it offers a multidisciplinary perspective encompassing biomedicine, biotechnology, and environmental applications. Furthermore, it includes analysis of green electrospinning's potential role in the circular bioeconomy and sustainable manufacturing. The review also discusses commercialization and tech transfer challenges in addition to research gaps. Finally, it provides an up-to-date overview of the state of green electrospinning research over the past 5 years. In summary, this review provides a more complete and forward-looking perspective compared to earlier reviews by covering the latest advances, applications, sustainability impacts, and commercial challenges that distinguish it as a unique contribution to the literature.

Electrospun nanofibers have gained immense interest due to their unique structure that mimics the nanoarchitecture of native extracellular matrices and tissues. Their highly porous structure combined with an enormous surface area-to-volume ratio makes them suitable for a variety of applications. In the biomedical field, electrospun nanofibers are widely used as tissue engineering scaffolds and platforms for controlled drug delivery [3–5]. They can be designed to meet the structural requirements of tissues and provide cues to enhance cell adhesion, proliferation and infiltration. Similarly, in the biotechnological domain, electrospun nanofibers have proven useful in areas such as biosensing, enzyme immobilization, and bioseparation processes due to their ability to mimic biological structures. However, traditional electrospinning methods employ harsh solvents and feedstocks, which raise sustainability concerns. The use of toxic chemicals can negatively impact both human health and the environment. It also limits the translation of electrospun materials for implantable and drug delivery applications due to biocompatibility issues [6,7]. To address these challenges, "green electrospinning" techniques have been developed that utilize environmentally friendly parameters and resources. These include the use of aqueous systems, biobased or biodegradable polymers, and alternative energy sources for processing. Transitioning to sustainable electrospinning practices is crucial to reduce the toxicity of manufacturing processes while preserving material functionality. Adopting green techniques produces nanofibers that are less cytotoxic and can readily degrade in vivo. This enhances their suitability for prolonged biomedical interventions [8–10]. This review aims to provide a comprehensive overview of various green electrospinning methods and evaluate their applications in biomedicine and biotechnology. It also discusses the challenges and future perspectives in the field of sustainable nanofiber manufacturing.

### 1.2. Importance of Green Electrospun Nanofiber Materials in Biomedicine and Biotechnology

Green electrospun nanofiber materials are increasingly important in biomedical and biotechnological applications due to their inherent advantages over conventionally produced nanofibers. The use of green techniques enables the production of nanofibers from nontoxic, renewable biomaterials such as collagen, gelatin, chitosan and cellulose. These biocompatible nanofibers can be safely implanted or incorporated into the human body without eliciting adverse reactions [11–13]. They are also biodegradable, allowing for degradation and absorption of the materials after serving their purpose. This makes green nanofibers highly suitable for applications involving tissue regeneration and drug delivery. In the field of biotechnology, green nanofibers allow for more sustainable fabrication of biosensors, affinity matrices and immobilization platforms. Their renewable nature reduces the environmental footprint and aligns well with the principles of green chemistry, engineering and manufacturing.

### 1.3. Purpose of the Review

The purpose of this review is to offer a comprehensive overview of the emerging field of green electrospinning for nanofiber material fabrication. In this endeavor, we delve into an in-depth analysis of various sustainable electrospinning techniques, shedding light on both their advantages and limitations. Our aim, in this comprehensive review, is to explore the representative applications of green electrospun nanofibers within the pivotal domains of biomedicine and biotechnology. These applications encompass the utilization of green electrospun nanofibers as indispensable components in tissue engineering scaffolds, as efficient carriers for controlled drug delivery, as sensitive biosensors, as reliable affinity matrices, and as versatile platforms for enzyme immobilization. Beyond their functional roles, we critically assess the environmental impacts and sustainability aspects of incorporating these green nanofibers into practical applications [14–16]. In addition to providing an extensive survey of the present landscape, we also delve into the current challenges that this field faces and lay out promising future perspectives for advancing green electrospun nanofibers. In embarking on this journey through our review, readers will attain a profound understanding of the potential role that sustainable nanofiber technologies can

play in developing safer and more environmentally friendly solutions for both medical and industrial applications.

## 2. Fundamentals

### 2.1. Basics of Electrospun Nanofiber Materials

Electrospun nanofiber materials refer to fibrous structures produced using electrospinning technology, with er-fiber diameters typically ranging from a few nanometers up to a few microns. Electrospinning is an er-fiber fabrication method in which a high-voltage electric field is used to draw very fine fibers from a polymer solution or melt. When the electric field is strong enough to overcome the surface tension of the liquid, a charged jet is ejected, and embedded fiber membranes are deposited on the target collector [17–19]. The fabricated nanofibers have high surface area-to-volume ratios and porous, nonwoven morphologies that make them suitable for a variety of applications. Their small fiber diameters and nanoscale pores are on the same scale as natural extracellular matrices, endowing them with desirable properties for mimicking native tissue structures. The electrospinning setup consists of three basic components: a syringe pump, a high-voltage supply, and a fiber collector. In the process, a polymer solution or melt is loaded into a syringe fitted to the pump. The tip of the syringe is connected to the high-voltage supply, which is adjusted to provide voltages typically ranging from 5 to 30 kV. The collector is earthed to create an electric field between the tip and the target. When the electric field overcomes surface tension forces at the tip, the polymer solution is drawn into a conical shape known as the Taylor cone. A charged jet is ejected from the tip as the electric repulsion overcomes surface tension. As the jet travels in the air gap towards the collector, solvent evaporation or solidification occurs, leaving behind thin solid fibers [2,20,21]. Process parameters such as the applied voltage, flow rate, tip–collector distance, and solution/melt properties regulate fibre formation and influence fiber diameter outcomes. In general, higher voltages tend to produce thinner fibers, as greater electric force-drawn jet undergoes greater elongation. However, very high voltages can cause fibers to split or become beaded. Increased flow rates result in thicker fibers deposited. At low flow rates, jet splitting occurs more easily. Longer tip–collector distances allow more time for solvent evaporation or solidification, but fibers may become beaded if the distance is too large. Shorter distances risk formation of non-woven mats without Taylor coning. Higher-viscosity solutions/melts produce thicker fibers compared to low-viscosity feeds. Concentration also impacts diameter. Ambient parameters like temperature and humidity influence solvent evaporation rate and fiber morphology. The fibers are collected as a nonwoven mesh on the grounded collector. The mechanism of electrospinning thus exploits electrostatic repulsion to produce fibers with diameters on the micron or nanoscale (Figure 1).

Table 1 summarizes natural polymers obtained from renewable plant and animal sources. Due to their natural origins, chitosan, collagen, gelatin, and cellulose have beneficial qualities, such as biodegradability, cell adhesion, and antibacterial activity. They have many uses in biomedicine, including drug delivery, tissue engineering, and the treatment of wounds [20,22]. Chitosan and cellulose are also suitable for biosensing and bioremediation in biotechnology. Efficiency arises from attributes such as biomimicry, sustainably sourced feedstocks and biodegradability. The key differences between the nanofiber materials include their source, properties, and applications. Firstly, the source refers to whether the materials are plant-based or animal-based. Additionally, the materials differ in their properties such as strength, bioactivity, and antimicrobial effects. Furthermore, the applications of the materials vary between uses like tissue scaffolds, drug delivery systems, and biosensing platforms [1,8,13]. The efficiency offered by these renewable biomaterials stems from several aspects. Firstly, their ability to biomimicry enhances cell interactions. Additionally, the biodegradable nature of the materials reduces long-term toxicity issues. Furthermore, sourcing the materials from renewable sources improves the overall sustainability of producing the nanofibers [6,17,22]. Biomedical roles focus on regenerative applications like implants, scaffolds and wound care. Biotechnological uses exploit properties like conduc-

tivity for biosensing and antimicrobial effects for environmental remediation. Overall, these renewable biomaterials enable greener nanofiber production [9,12,18].

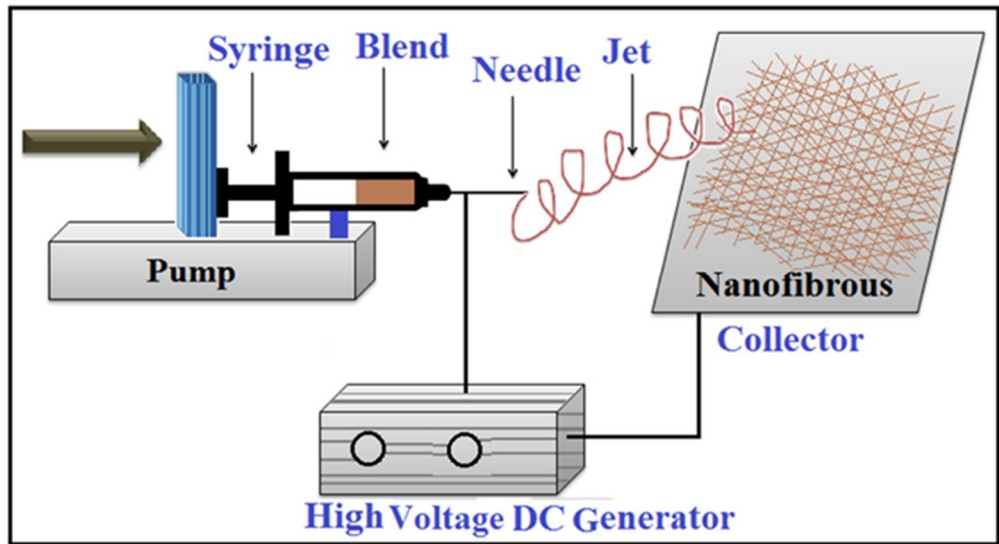

**Figure 1.** A diagram illustrating the schematic of the electrospinning process utilized for the production of nanofibers [2].

**Table 1.** Green electrospun nanofiber materials: Natural polymers from renewable sources.

| Nanofiber Type | Basic Properties | Biomedical Properties | Biotechnological Properties | Ref. |
|---|---|---|---|---|
| Cellulose | Renewable from plants/wood, Biocompatible, biodegradable, High strength, flexibility | Tissue scaffolds, Wound dressings, Vascular grafts | Biosensors, Biocatalyst, immobilization | [1–5] |
| Collagen | Obtained from animal tissues, Contains bioactive peptides, Supports cell adhesion | Skin/bone regeneration, Hernia repair meshes, Nerve conduits | N/A | [11–16] |
| Gelatin | Collagen derivative, Tunable degradation, Low immunogenicity | Drug delivery, Wound healing, Soft tissue scaffolds | Enzyme immobilization, Affinity membranes | [6–10] |
| Chitosan | From shrimp shells, Antimicrobial activity, Biocompatible | Wound dressings, Tissue engineering, Bone regeneration | Biosensing, Bioremediation | [17–22] |

Table 2 features synthetic biopolymers produced through fermentation of renewable feedstocks rather than fossil fuels. PLA and PBS are biodegradable thermoplastics with tunable degradation and mechanical properties such as strength and flexibility, respectively. These traits enable their use as implants, sutures, medical devices and drug carriers. While biomedical roles are established, biotechnological uses are relatively unexplored for these biomass-derived polymers. The key differences between the synthetic biodegradable polymer nanofiber materials include their monomer source and material properties. Firstly, the monomer source refers to whether the polymer is derived from starch or succinic acid. Additionally, the materials differ in their flexibility versus strength properties [23,24]. The efficiency offered by these materials stems from several aspects. Firstly, utilizing a renewable biomass source as the monomer improves the overall sustainability. Additionally, the tunable biodegradation reduces long-term toxicity issues [25–37]. For biomedical applications, PLAs have been utilized as bioresorbable implants, sutures, and tissue engineering scaffolds. Meanwhile, PBS is employed in medical devices and as drug carriers. Overall, these bioplastics are produced through fermentation of renewable resources rather than

petrochemical feedstocks. Their biodegradability and mechanical properties make them suitable for applications requiring resorption like sutures and implants. However, their uses in biotechnology have not been extensively explored [25,38].

**Table 2.** Green electrospun nanofiber materials: Synthetic biodegradable polymers from biomass.

| Polymer Nanofiber Type | Source | Basic Properties | Biomedical Properties | Biotechnological Properties | Ref. |
|---|---|---|---|---|---|
| Poly(lactic acid) (PLA) | Plant starch/sugar fermentation | Thermoplastic, Tunable degradation rate, High strength/elasticity | Implants, Sutures/meshes, Tissue engineering scaffolds | N/A | [23–29] |
| Poly(butylene succinate) (PBS) | Succinic acid from plants/microbes | Flexibility, Impact resistance, Biodegradation in soil/compost | Implants, Medical devices, Drug delivery carriers | N/A | [30–33] |

Table 3 presents composite nanofibers that combine two or more green materials. Blends such as cellulose/chitosan and gelatin/hydroxyapatite impart multifunctionality with features such as bioactivity, drug loading and enhanced mechanics. These composites focus on applications in tissue engineering, wound healing and controlled drug delivery for biomedicine [1–5]. The cellulose/chitosan system also shows potential as a biosensor in biotechnology. Across tables, key differences are noted in the polymer sources, inherent properties and resulting applications dictated by the renewable monomers and compositional flexibility. Biomedical roles predominantly serve regeneration and therapeutics [6–10]. The efficiency offered by these composite green nanofibers stems from several aspects. Firstly, they are able to biomimic native tissues. Additionally, they enable sustained drug release. Furthermore, the composites exhibit improved mechanical properties when compared to single component fibers [39–41]. Their applications include tissue engineering and regeneration. They also find uses in wound care applications. Furthermore, the composites serve as controlled drug delivery systems. By combining renewable polymers and compounds, the materials provide multifunctionality like bioactivity, drug loading capacity, and enhanced mechanics. The biomedical roles of the composites primarily focus on implants and therapeutics [42–44].

**Table 3.** Green electrospun nanofiber materials: Composite green nanofibers.

| Composite | Characteristics | Biomedical Properties | Biotechnological Properties | Ref. |
|---|---|---|---|---|
| Cellulose/Chitosan | Biocompatible, renewable polymers | Tissue engineering scaffolds, Wound dressings | Biosensors, Affinity membranes | [16,39–42] |
| Gelatin/Hydroxyapatite | Mimics bone composition | Bone regeneration, Oral implants | N/A | [43–48] |
| Curcumin/Gelatin | Natural anti-inflammatory drug | Controlled drug delivery, Wound healing | N/A | [49–55] |

### 2.2. Green Electrospinning Methods

Green electrospinning represents a more sustainable approach to the production of electrospun nanofiber materials, with the primary objective of mitigating the environmental and health concerns associated with conventional methods. Traditional electrospinning relies on harsh organic solvents, high voltages, and nondegradable polymers, which give rise to a host of challenges such as the safety hazards posed by toxic solvents, limitations in clinical applications due to cytotoxic components, and the environmental impact of nonrenewable materials [19,41,44]. Green electrospinning addresses these limitations through several innovative strategies. Firstly, it substitutes aqueous systems for organic solvents, effectively eliminating toxicity concerns. Additionally, this approach explores

solvent-free techniques to eliminate hazardous chemicals from the process and focuses on using biodegradable, renewable polymers sourced from natural materials. Moreover, it investigates the utilization of sustainable energy sources, such as sunlight, to power the electrospinning process. Transitioning to these green electrospinning techniques yields significant benefits. It overcomes regulatory barriers by producing nanofibers that are compatible with biological systems, enabling their prolonged use in biomedical applications without adverse reactions. Furthermore, it aligns with the principles of green chemistry by utilizing eco-friendly materials and contributes to the reduction in the ecological footprint associated with nanofiber manufacturing. In summary, green electrospinning is driven by the ambition to create safer, non-toxic nanofibers while retaining their functionality, making them suitable for both biomedical translation and industrial-scale production. This transition towards more sustainable practices holds promise for a greener and healthier future in nanofiber technology.

Various green techniques have been developed that offer more sustainable alternatives to conventional electrospinning, which relies on toxic solvents and high-energy inputs. As summarized in Table 4, these methods can eliminate solvent usage, reduce energy consumption, enable renewable production routes, and utilize a wider range of materials. The aqueous, emulsion, and melt electrospinning techniques process polymers without harsh chemicals by employing water or thermal energy. This removes toxicity concerns while maintaining functionality. Near-field and centrifugal spinning also address the energy intensiveness issue through mechanical or lower electrical field approaches. Bacterial nanocellulose and solar electrospinning showcase truly renewable options [21,44,53]. Bacterial culture facilitates the in situ growth of cellulose nanofibers from biomass. Solar power directly harnesses sunlight to drive the electrospinning process without any electricity. Key benefits offered by these greener methods include solvent avoidance, applicability to diverse materials, reduced processing demands, and self-powered or biomass-based manufacturing. These enhancements help realize the principles of green chemistry and engineering for electrospun materials [53–55]. Overall sustainability is improved compared to conventional techniques through various innovations.

**Table 4.** Summary of the key green electrospinning techniques [21,44,53–55].

| Technique | Description | Benefits |
|---|---|---|
| Aqueous Electrospinning | Uses water as a solvent instead of organic chemicals. Suitable for water-soluble polymers like collagen, gelatin, chitosan. | Eliminates the use of toxic organic solvents. |
| Emulsion Electrospinning | Involves water-in-oil or oil-in-water emulsions for insoluble polymers. | Provides benefits of aqueous systems while maintaining material compatibility. |
| Melt Electrospinning | Feeds polymers in melt/semisolid form directly through the nozzle without solvents. | Applicable to thermoresponsive polymers like PLA, PCL. Solvent-free process. |
| Near-Field Electrospinning | Performs electrospinning at short, 1–5 mm tip-collector distances and low voltages (<5 kV). | Requires lower electric field strengths. |
| Centrifugal Spinning | Uses centrifugal rather than electrostatic force to form fibers. | No chemical exposure, high voltages or expensive equipment needed. |
| Bacterial Nanocellulose Spinning | Facilitates in situ growth of nanocellulose hydrogels on a rotating surface via bacterial culture. | Completely biomass-derived and renewable fiber production. |
| Solar Electrospinning | Replaces high voltage supply with photovoltaic cells powered by sunlight. | Highly sustainable process with no electrical energy requirement. |

## 3. Applications in Biomedicine and Biotechnology

### 3.1. Tissue Engineering and Regeneration

### 3.1.1. Tissue Engineering Scaffolds

Summarizing the key points discussed in the context of tissue engineering and regeneration applications of green electrospun nanofiber materials (as detailed in Table 5) [55–57], it becomes evident that green electrospun nanofibers hold immense promise in the realm of tissue engineering and medical device development. This potential is attributed to their unique ability to closely mimic the structural properties of natural extracellular matrices. These nanofibers possess several crucial properties, including nanoscale fiber diameters, high porosity, tunable biodegradability, and impressive mechanical strength. Notably, biopolymers such as cellulose, collagen, and gelatin emerge as ideal candidates for scaffold design, with their renewable nature supporting tissue regeneration. Functioning as scaffolds, these fibers effectively facilitate cell infiltration and nutrient diffusion while gradually degrading to accommodate the replacement by new tissue. Their applications extend to various areas, including scaffolds for skin, bone, and cartilage repair. In the context of vascular grafts, collagen-containing fibers closely replicate the mechanical function of blood vessels, while tunable gelatin/PLA blends offer a balance between strength and remodelling. The cell-interactive surfaces of these fibers encourage cellularization, enhancing their functionality. Key characteristics such as sufficient pore size, matrix mimicry, and the provision of dynamic mechanical cues play a pivotal role in enabling successful tissue regeneration. Importantly, the use of renewable resources in constructing these scaffolds aligns with sustainable healing outcomes, contributing to a more environmentally responsible approach. Future research endeavors aim to optimize scaffold microarchitecture, degradation kinetics, and functionalization with cues that stimulate targeted regeneration. Given their biocompatibility and the promise they hold, green electrospun nanofibers are poised to emerge as next-generation tissue engineering platforms, offering innovative solutions for the field of regenerative medicine. The key characteristics of the green electrospun nanofiber materials for use as tissue engineering scaffolds include their nanostructure and high porosity, which enable cell infiltration. Additionally, the biodegradable nature of the fibers allows for dynamic scaffold properties over time as the tissues regenerate. The green fibers employ renewable sources, which permits regenerating tissues safely and sustainably. Future work in this area may aim to optimize aspects like mechanical cues, functional cues, and degradation kinetics to further improve the regeneration observed when using these scaffolds [56–71].

**Table 5.** Green electrospun nanofiber materials in Tissue Engineering Scaffolds.

| Material | Basic Details | Main Responsible | Greenness | Future Suggestions | Ref. |
|---|---|---|---|---|---|
| Cellulose | Nanostructure, >90% porosity, degradation over months | Skin, bone regeneration | Renewable source, biodegradable | Functionalization with growth factors, mechanical strengthening | [56–60] |
| Collagen | Diameter 50–500 nm, 80–90% porosity, degradation over months | Skin, cartilage regeneration | Biomimicking ECM, biodegradable | Angiogenic/osteogenic functionalization, controlled degradation | [61–66] |
| Gelatin | Diameter 100–300 nm, 70–85% porosity, degradation over weeks | Skin, nerve regeneration | Biodegradable, processed from collagen | Crosslinking for strength–porosity control, drug release studies | [67–71] |

Figure 2 examines tendon fibroblast orientation on aligned vs random PLGA fibers. At Days 3 and 7, staining showed cell elongation and aligned F-actin on oriented fibers (A, C, D), indicating contact guidance. Random fibers lacked cues, offering irregular spreading (B). This demonstrates that fiber alignment influences cytoskeletal organisation and orientation over time [72]. Figure 3 compares BM-HSC capture on substrates. Rounded cells captured

on E-selectin-coated collagen-blended PLGA nanofibers (B), but none on tissue culture polystyrene (A). This indicates that the bioactive coating mediated initial attachment [73].

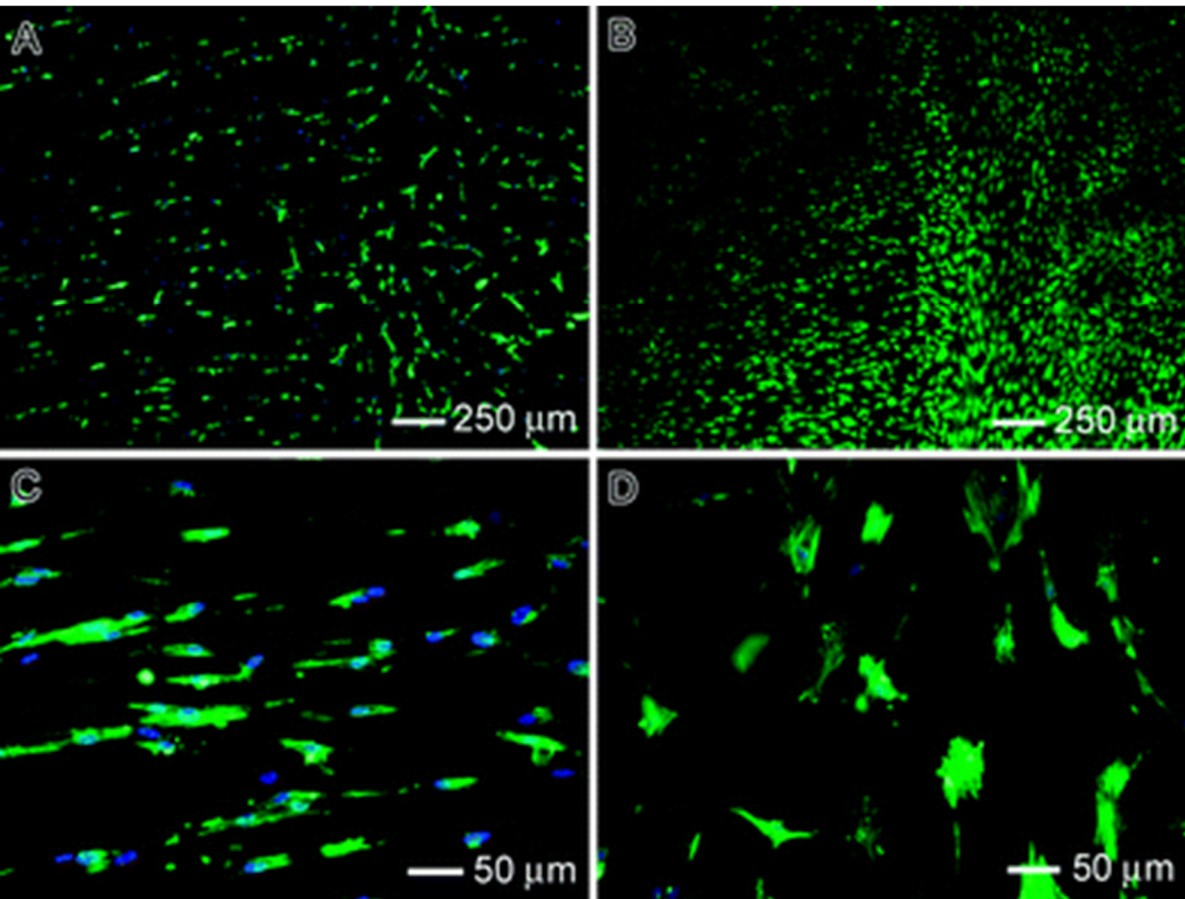

**Figure 2.** Shape of tendon fibroblast cells on aligned versus disorderly PLGA (50:50) fiber frameworks: (**A**,**B**) subsequent to retaining for 3 and 7 days, individually. (**C**,**D**) Magnified perspectives of (**A**). In (**A**,**C**,**D**), F-actin and cores of the cells were dyed with FITC–phalloidin and DAPI, individually, in green and dim hues. The cells in (**B**) were stained with FDA in green shading [72].

### 3.1.2. Vascular Grafts

Green electrospun nanofibers show potential as small-diameter vascular grafts for blood vessel reconstruction (Table 6). Compliant fibers resembling the mechanical properties of native vessels improve upon synthetic graft materials currently used. Biopolymers such as collagen/silk mimic the extracellular matrix composition of arteries. Composite fibers allow tuning of the strength degradation balance over time, with gelatin/PLLA evaluated for venous applications (Figure 4). Importantly, materials such as cellulose/gelatin produce nonthrombogenic fibre surfaces conducive to endothelial cell infiltration and remodelling of the neovessel. This cell-interactive property addresses the failures of nondegrading synthetic polymers. Key characteristics of graft materials include compliance matching surrounding tissue, controlled biodegradation, and nonthrombogenic function supporting cellular remodelling into functional blood vessels. By utilizing renewable biomaterials as fibre composition, these green electrospun constructs demonstrate more sustainable alternatives to petroleum-derived vascular conduits. Future work will continue to optimize the biodegradation kinetics, and proangiogenic functionalization strategies of nanfibers. Overall, green nanofibers present promising next-generation platforms. The key characteristics required for the green electrospun nanofiber materials to be utilized as vascular grafts include compliance similar to native blood vessels. Additionally, the materials must exhibit nonthrombogenicity to safely contact blood flowing through the grafts.

Another important characteristic is tunable strength degradation properties over time. Finally, the ability to support cell infiltration and remodeling capacity is crucial [73–85].

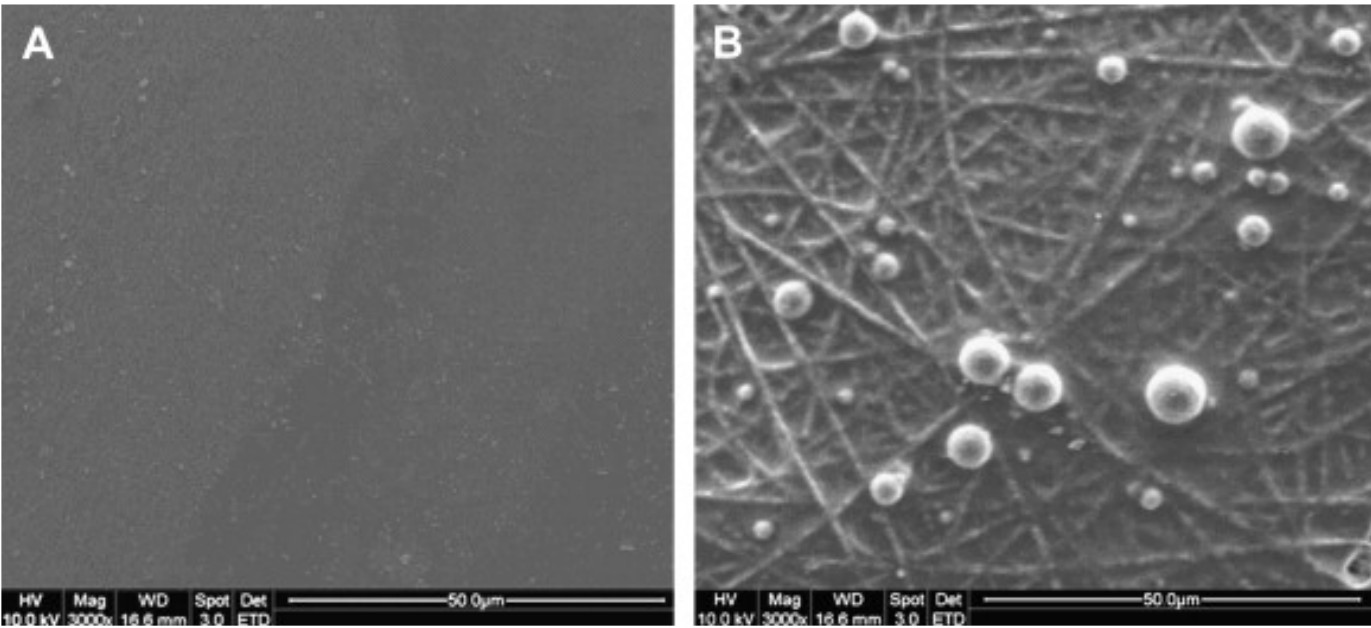

**Figure 3.** Capture of BM-HSCs by different substrates after 30 min of incubation. ((**A**) no BM-HSCs captured on TCP, (**B**) rounded morphology of BM-HSCs captured on E-selectin-coated collagen-blended PLGA NFS) [73].

**Table 6.** Green electrospun nanofiber materials in Vascular Grafts.

| Material | Basic Details | Main Responsible | Greenness | Future Suggestions | Ref. |
|---|---|---|---|---|---|
| Collagen/Silk | Diameter 50–500 nm, resembles veins/arteries | Arterial grafts | Biomimicking ECM, biodegradable | Angiogenesis cues, mechanical properties | [74–78] |
| Gelatin/PLLA | Tunable strength-degradation, moderate compliance | Venous grafts | Biodegradable polymers | Strengthening, nonthrombogenic surface | [79–83] |
| Cellulose/Gelatin | High porosity, nonthrombogenic, remodelling | Arterial/venous grafts | Renewable materials | Mechanical properties, controlled degradation | [84–87] |

### 3.2. Controlled Drug Delivery

#### 3.2.1. Antibiotic and Anticancer Delivery

Biopolymer carriers fabricated using green electrospinning methods can provide sustained drug release for improved therapies. Gelatin fibers release norfloxacin steadily for 3 weeks from their hydrophilic matrix, maintaining antibiotic levels to prevent wound infections longer than instant doses. Chitosan nanofibers also sustain doxorubicin release for 4 weeks to effectively treat cancers locally via their drug-complexing positive charges [87–90]. Targeting agents such as hyaluronic acid, antibodies and aptamers functionalize antibiotic/chemo-loaded scaffolds, directing delivery selectively to sites such as inflamed tissues or tumors. Compared to immediate bolus dosing, consistent drug levels from these targeted, biodegrading platforms reduce side effects while maintaining potency. Application in cancer regression and wound healing is prominent. As renewable polymers amenable to diverse processing, they offer green alternatives to permanent drug depots [91–94]. Control over fibre properties and drug interactions tunes customizable release profiles [95–97]. Optimizing targeted formulations and correlating fabrication to

pharmacokinetics can advance clinical translation. Advanced constructs with multidrug, sequential release may achieve superior therapeutic outcomes. Overall, these sustainable platforms demonstrate promising personalized therapies through safe, local drug administration tunable for each application.

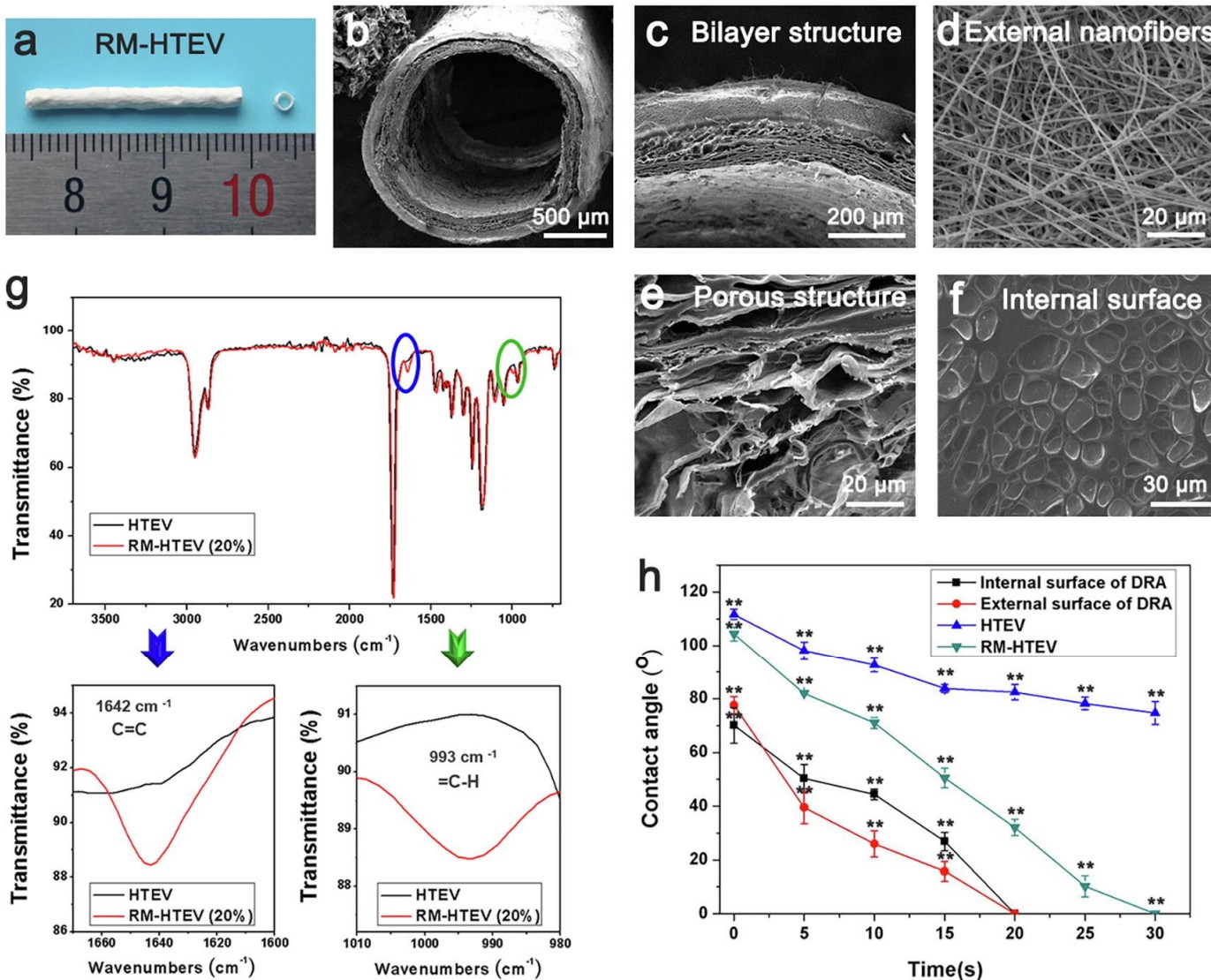

**Figure 4.** Characterization of reconstituted muscle tissue engineered vascular implant (RM−HTEV): (**a**) Visual image of RM−HTEV from above view and section view. (**b**−**f**) Scanning electron microscope (SEM) photographs displaying the neatly organized passageway ((**b**), measuring scale = 500 μm) of RM−HTEV, comprising the double−layered structure ((**c**), measuring scale = 200 μm), dense exterior RM−loaded polycaprolactone (PCL) nanofibers ((**d**), measuring scale = 20 μm) and porous interior decellularized rat aorta (**e**), measuring scale = 20 μm; (**f**), measuring scale = 30 μm). (**g**) Fourier transform infrared spectroscopy (FTIR) examination showing the profitable fabrication of the RM−loaded PCL nanofibrous layer of RM−HTEV. (**h**) Contact point values of the internal surface of DRA, external surface of DRA, external surface of HTEV and external surface of RM−HTEV at di−verse times (value ± SD), ** $p < 0.01$ versus the other groupings at the equivalent time point. The inner and outer surfaces of DRA displayed good wettability. Moreover, the surface of RM−HTEV demonstrated significantly heightened wettability compared to the hydrophobic exterior of the pure PCL nanofiber layer [85].

### 3.2.2. Gene Delivery

Green electrospinning shows potential for producing nonviral gene delivery vectors. Chitosan, gelatin and cellulose nanofibers have positively charged backbones enabling complexation with therapeutic nucleic acids. This protects genetic payloads until cellular uptake. For example, gelatin/chitosan fibers condense plasmid DNA and release it gradually over 7 days from a scaffold. Nontoxic carriers avoid the safety issues of viral vectors. Tissue regeneration applications include growth factor gene-embedded scaffolds to stimulate wound repair or angiogenesis [98–101]. Sustained nucleic acid delivery mimics viral transfection efficiencies. Biopolymer biocompatibility and fibre nanoarchitecture facilitate scaffold-based delivery locally. Fibre hydrophobicity/porosity tunes protection and release kinetics of delicate cargo. Future work aims to enhance transfection and spatiotemporal control. For instance, targeting ligands may enable regeneration of specific organ systems through localized gene therapy [102–104]. Overall, these green nanocarriers show potential as safe, nonviral gene delivery tools for advanced tissue engineering applications requiring DNA/RNA payloads. As biodegradable and renewable materials, they fulfil the principles of green biomaterial design.

### 3.2.3. Hemostatic Dressing

Green electrospun fibers form hemostatic wound dressings that control hemorrhage and prevent infection. Cellulose and collagen fibers accelerate clotting via mechanical absorption of blood and platelet interactions. Their porous structures localize hemostatic agents at wound beds. Silver nanoparticle- or chlorhexidine-loaded gelatin nanofibers release antimicrobials in controlled bursts to minimize infection risks as wounds heal. Efficacy as hemostats shortens the time to achieve bleeding cessation compared to gauze. As biopolymers, materials avoid biological incompatibility of synthetics. Future work optimizing loading levels and tailored release profiles would provide minimum bacteriostasis over suturing and healing periods [105–108]. Advanced 3D fibre architectures may simulate complex tissue geometries to homogeneously interact with irregular wound surfaces for rapid, infection-free healing. These functionalized green wound dressings show potential through combined hemostatic and antimicrobial functions accomplished safely using renewable biomaterials.

### 3.2.4. Burn Wound Dressing

Green electrospun fibers create burn dressings that efficiently absorb the exudate while maintaining a moist environment. Nanoscale gelatin and cellulose dressings rapidly intake fluids via a high surface area yet remain permeable, preventing further tissue damage from contact with dry surfaces. Moisture-managing abilities accelerate re-epithelialization compared to air-drying methods. Hydrophilic fibers absorb excess wound fluid for patient comfort. Biocompatible materials eliminate risks from synthetic polymer sensitivities in open burn injuries. Biodegradable fibers do not require removal, avoiding painful changes. Future work enhancing mechanical properties and developing intelligent dressings responsive to wound conditions may optimize healing trajectories. Advanced formulations incorporating regrowth factors better emulate the ECM microenvironment to augment the tissue regeneration capacity of green wound care products [109–112]. Overall, these green fibre dressings advance burn wound healing through effective exudate management and provide a moist, protected surface using nonirritating, renewable materials (Figure 5).

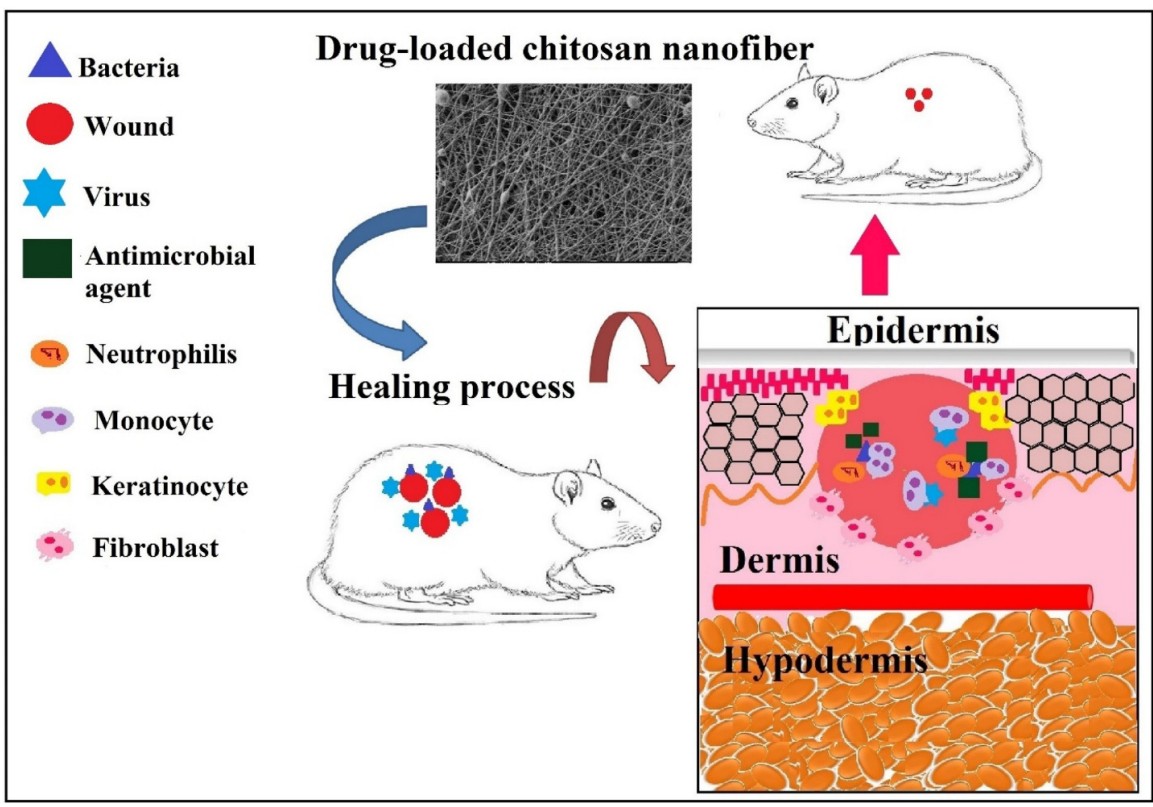

**Figure 5.** Representation of the healing process in a wound rat model.

### 3.2.5. Sutures/Medical Textiles

Green electrospinning produces fibrous meshes and constructs for soft tissue repair. Collagen, silk and polyester meshes act as biodegradable surgical sutures with controllable strength retention over healing periods. Tuned degradation matches native ECM turnover. Gelatin/PLGA nanofibrous meshes serve as dermal or pericardial scaffolds. Porous structures encourage cellular infiltration without adverse reactions to polymers. These renewable textile constructs replace petroleum-derived nonabsorbable meshes and sutures. Biocompatibility eliminates foreign body responses. Future work optimizing fibre alignment and mechanical properties may develop anatomical replacements such as ligaments or tendons. Cardiovascular patches may incorporate endothelial cells on efficiently vascularizing fibre surfaces for rapid tissue regeneration [113–116]. Overall, green electrospun medical textiles provide safe, sustainable alternatives to synthetic materials in soft tissue applications through timeframe-appropriate properties and elimination of synthetic polymer sensitivities.

### 3.2.6. Implants

Biodegradable green electrospun implants balance mechanical integrity with degradation. Composite fibers combining polycaprolactone with collagen show porosities encouraging osteoblast ingrowth while tailoring implant stability over months as bone regenerates. Gelatin/hydroxyapatite scaffolds support new cartilage formation with calcium/phosphate reinforcement degrading as tissue matures [117–121]. Future directives include optimizing fibre architectures for anatomical congruency and developing intelligent implants incorporating cellular communication for accelerated healing. Printed conductive networks pairing synthetic resorption with biosignaling may stimulate specific tissue lineages [122–125]. By replacing permanent metal/plastic implants, renewable fibre composites provide standardized resorption matched to tissue reconstruction in bone/cartilage regeneration. As bioresorbing alternatives utilizing sustainable materials, green electrospun implants aim to restore natural function while avoiding chronic foreign body effects.

Electrospun nanofibers have advantages over macro-scale fibers like spinning/drawing due to their nanoscale diameters, which provide high surface area/volume enabling higher drug loading and more efficient release versus microscale fibers. Small diameters and high porosity also better mimic tissue extracellular matrix than larger fibers, enhancing cell interactions. Fibers can be directly produced from polymers during electrospinning without downstream processing like spinning, simplifying manufacturing. A wide variety of natural/synthetic polymers allows tuning properties for applications. Incorporating nanoparticles into electrospun fibers enables controlled release comparable/superior to micelles/nanoparticles through homogeneous encapsulation during single-step electrospinning for multi-functional payloads, enabling longer sustained/stimuli-responsive release than diffusion micelles/nanoparticles while nanoparticles protect cargos from issues faced by colloids.

### 3.3. Applications in Biotechnology

#### 3.3.1. Biosensors

#### Enzyme Immobilization

Green electrospinning produces nanofiber-based biosensors through enzyme immobilization. Cellulose and chitosan fibers entrap glucose oxidase and cholesterol oxidase, respectively, for detection. Nanoscale templating preserves enzyme activity through mild processing. Fibre geometry and reactive surface functionalities such as amine/carboxylate groups covalently tether biomolecules [126–130]. Enzyme-embedded conductive composite fibers electrically translate biological recognition into quantifiable signals. Applications include point-of-care testing and food analyses. Renewable materials avoid petrochemically derived sensor substrates, offering sustainable alternatives. Future work optimizing enzyme loadings and electrical conductivity would achieve faster, lower-cost diagnoses [131–135]. Core-shell and 3D nanostructure designs that spatially separate recognition/transmission elements present new detection mechanisms. Overall, these green sensing platforms demonstrate biocompatible, biodegradable matrices for developing applications in bioprocess monitoring and analytical biotechnology.

#### Immunosensors

Green electrospun nanofibers develop immunosensor platforms by functionalizing fibre surfaces. Collagen fibers immobilize capture antibodies through coordination bonds with secondary antibodies, facilitating analyte detection via antigen–antibody interactions. Sensitive, renewable gelatin/chitosan composites coprint antibodies and interdigitated electrodes onto a single substrate. Functionalized surfaces concentrate specific molecule recognition for amplified signals transduced electrically [136–141]. Applications involve disease diagnostics and food/water safety testing more quickly/affordably than laboratory techniques. Biopolymer surfaces stabilize fragile biomolecules while allowing molecular access. Future work may develop multiplexed detection of pathogen/toxin panels or optimization of the Optifin Reader for future personalized rapid diagnostics [142–147]. Overall, these green recognition interfaces coupled with miniaturized electronics offer biocompatible, integrated immunosensor systems for applications in point-of-care testing through portable device formats.

#### Cell Growth Substrates

Green electrospun nanofibers act as cell culture supports, controlling the microenvironment. Gelatin scaffolds with diverse fibre sizes/densities influence cell morphology, proliferation and differentiation signals. Amine-functionalized polyvinyl alcohol nanofibers covalently link cell adhesive peptides to interact with integrin receptors and modulate behavior. The 3D nanotopographies better mimic in vivo conditions compared to standard tissue culture plates. Applications involve constructing functional tissues such as skin and bone from expanded cell populations on renewable surfaces. Biodegradability allows recycling cell-seeded constructs as implantable scaffolds. Future work optimizing

material-specific cues and developing complex 3D constructs could advance organ-on-a-chip platforms for more predictive toxicity/efficacy modelling [148–150]. Overall, tunable green electrospun substrates deliver customizable microenvironments, improving basic research methods through naturally derived engineered tissues.

Tissue Modelling

Green electrospun nanostructures aid in the construction of tissue models recreating organ architecture in vitro. Multilayered gelatin–collagen stacks incorporate combinations of endothelial cells, fibroblasts and keratinocytes on adjacent fibre sheets to emulate skin tissue barriers. Placed between fluid chambers, layered open-pore nanofiber membranes model intestinal microenvironments by culturing gut and blood endothelial populations on opposite surfaces. Biomimetic 3D topographies and tissue-selective cell seeding using renewable materials better simulate in vivo complex tissue interfaces than standard 2D cocultures [151–156]. Applications involve developing organ models for drug testing and regenerative therapies. Biodegradability facilitates downstream scaffold and implant development using tissue-engineered constructs. Future work may refine tissue structure fidelity and incorporate multiple cell and fluidic interaction dynamics for advanced organ and whole body-on-a-chip platforms. In summary, these controlled 3D tissue models advance disease and toxicology studies, replacing animal testing.

### 3.4. Environmental Remediation

### 3.4.1. Water Purification

Green electrospun nanofiber membranes enable contaminated water treatment through adsorption and filtration. Cellulose nanofiber mats effectively absorb pollutants such as dyes and heavy metals from wastewater via their high surface area and reactive hydroxyl groups. Composite chitosan–gellan gum nanowebs functionalized with metal-chelating groups capture specific toxic ions during filtration. High porosity maintains throughput without the application of renewable biomass and provides an eco-friendly alternative to petroleum-based purification methods [157–160]. Future work optimizing membrane thickness and multilayer designs could realize real-world applications addressing challenges such as desalination or industrial effluent remediation. In-line sensor integration would offer automated filtration monitoring as well [161–165]. Overall, these sustainable nanofibrous platforms present a green solution for water purification applications through nanomaterial-enabled adsorptive and sieving mechanisms.

### 3.4.2. Air Filters

Green electrospun nanofibers are effective air filtration media for industrial and medical applications. Nonwoven gelatin/PVA mats coated with antibacterial silver nanoparticles capture airborne pathogens in HVAC systems or hospitals. Loose nanofiber networks allow high flow rates while trapping particles/microbes >100 nm via tortuous diffusion paths. Composite chitosan/cellulose filters chemisorb industrial pollutants such as volatile organics and heavy metals from factory fumes [166–170]. As renewable materials, these filters offer sustainable alternatives to synthetic options such as HEPA filters. Optimizing fibre charges and developing self-cleaning properties may realize real-world industrial stack gas cleaning and sterile lab environments [171–175]. In summary, green electrospun air filters show promise for environmental and occupational applications through renewable, high-efficiency filtration of microbiological and chemical air pollutants.

### 3.4.3. Controlled Release Fertilizers/Pesticides

Green electrospinning develops agrochemical carriers that provide sustained nutrient/crop protection. Gelatin/clay composite fibers encapsulate fertilizer salts, releasing them gradually as gelatin degrades. This matches plant uptake over growth cycles. Biopesticide-loaded cellulose nanofibers incorporated as mesh barriers maintain insecticide levels for weeks to deter pests. Precisely metering agrochemicals optimizes yields while

minimizing pollution from runoff. Renewable polymers substitute for nonbiodegradable formulations [1,176–179]. Controlled release also reduces application frequency/amounts applied. Future formulations tailoring release profiles address diverse soil/climate conditions. In summary, these green platforms balance agricultural productivity with environmental sustainability through customized nutrient/substance delivery from degradable fibre networks.

### 3.4.4. Food Packaging

Green electrospun nanofibers develop active food packaging with antimicrobial properties. Zinc oxide nanoparticle-loaded polyvinyl alcohol coatings on cellulose fibre substrates inhibit bacterial growth on perishable goods. Electrospun gelatin fibers incorporating grapefruit extract not only kill microbes, but also prevent resuspension through a protective physical network. These properties enhance shelf life through multifunctional antioxidant and antibacterial actions compared to passive storage containers [180–184]. Renewable materials replace conventional plastic packaging. Degradability ensures that materials do not persist in the environment upon disposal. Customization with essential oils tailored for specific produce types could optimize microbial inhibition effects [185–188]. In summary, green active nanofiber coatings provide a sustainable solution to reduce foodborne illness through natural, biodegradable preservation technologies.

### 3.4.5. Cell Encapsulation

Green electrospinning develops cell encapsulation platforms for therapeutic applications. Alginate microcapsules codeliver islet cells within chitosan–gelatin nanofiber coatings. Pore sizes allow metabolic exchange, while barriers prevent immune cell infiltration and transplant rejection. Encapsulation within collagen-HA fibers as cell-interactive hydrogels maintains the viability of encapsulated hepatocytes through innate signaling cues. Mild biomaterials advance cell therapies by preventing immune destruction without pharmaceuticals. Future work optimizing capsule mechanical/mass transport properties using advanced bioinking may scale production for treating diabetes or liver diseases [189–192]. These renewable encapsulation systems offer biocompatible, localized cell immunoprotection through customizable biomaterial–cell interactions.

### 3.4.6. Prodrug Activation

Green electrospun scaffolds integrate synthetic biology for controlled multidrug release. Polyester fibers coencapsulate enzyme-expressing E. coli and inactive prodrug conjugates. At infection sites, bacteria metabolize conjugates, activating drugs in sustained bursts. Conductive mixed-ligand hydrogels localize drug-resistant bacteria and anodes, killing cells through reactive oxygen species while releasing drugs [192–198]. Such living materials bypass many drug resistance challenges by coupling enzymatic activation with synergistic therapies. Biocompatible materials facilitate implantation of these engineered bacterial systems. Future work optimizing signals and population dynamics may realize advanced in vivo diagnostics and therapeutics. In summary, these platforms demonstrate the potential at the nexus of green materials and synthetic biology for developing personalized prodrug delivery strategies.

### 3.4.7. Fiber Diameter in Electrospinning

Fiber diameter is integral to the process and concept of electrospinning itself. Key factors like applied voltage, flow rate, tip-to-collector distance, and polymer properties have direct impacts on the attained diameter. For example, collagen fibers can range from 50 to 500 nm while PLA can achieve diameters from 200 nm up to 5 μm, varying over orders of magnitude simply by tweaking these parameters. This ability to engineer materials down to the nanoscale distinguishes electrospun fibers from those made by conventional microfiber production methods. Replicating naturally occurring dimensions, like collagen fibrils in the 50–500 nm range, is what underpins its biomedical relevance. Most fundamentally, it is

the charge-induced bending instabilities imparted to the polymer jet as it travels under an applied electric field that enable the establishment of exceedingly thin fibers. Fiber diameter is thus a direct and defining output of the electrospinning process, governed by the very mechanism enabling diameters to be precisely controlled within the distinctive nanoscale regime. This capacity for genuine nanofiber generation is what endows electrospinning with its unique character and multidimensional application landscape compared to other fiber spinning techniques. In this way, fiber diameter is entirely intrinsic to the technology rather than an extrinsic consideration.

## 4. Environmental Impact and Sustainability

*Discussion on the Environmental Footprint of Green Electrospun Nanofiber Materials*

Conventional electrospinning relies on toxic chemicals and energy-intensive processes, contributing to a large carbon footprint and ecological damage. However, green electrospinning techniques aim to lower this environmental impact through various approaches [58,59,61]. Biopolymers such as cellulose, gelatin and chitosan can be easily sourced from agricultural/marine waste. This provides an abundant, sustainable supply of raw materials. Removing organic solvents from the production process prevents toxicity issues associated with their manufacture, use and disposal. It also reduces regulatory hurdles for applications. Technologies such as solar, centrifugal and near-field electrospinning markedly decrease electricity usage. This cuts carbon emissions compared to traditional methods' high-voltage demands. Certain processes couple nanofiber fabrication with wastewater treatment or $CO_2$ sequestration. This turns waste into a resource, further improving eco-efficiency. Biodegradable nanofibers from green synthesis can re-enter the biosphere after use, closing the loop on material flows. Overall, green electrospinning strategies minimize dependence on nonrenewables, cut pollution, utilize agricultural/industrial side streams, and promote more circular material economies. This lowers the environmental impacts of nanomanufacturing.

Several sustainability aspects of green electrospinning technologies were identified in the literature. In terms of resource efficiency, green electrospinning utilizes renewable biomass sources that do not compete with food production. Additionally, it recovers materials from waste streams such as lignocellulose and food processing residues. Furthermore, it implements closed-loop manufacturing by designing for disassembly and remanufacturing [58–60]. Regarding cleaner production, green electrospinning eliminates toxic organic solvent emissions and wastewater effluents. It also incorporates wastewater treatment or degassing of volatile compounds. Additionally, it utilizes aqueous, solvent-free and mechanistic fabrication routes. In terms of energy efficiency, solar, centrifugal and near-field methods utilized in green electrospinning drastically cut electrical energy usage. Furthermore, the process may be powered through renewable energy sources such as solar and wind. It also integrates with sustainable energy generation applications. Concerning environmental performance, green electrospinning produces fully biodegradable nanofibers that re-enter the biosphere. Additionally, it sequesters carbon through algal and bacterial cultures used in production. The approach also assesses lifecycle impacts through analysis of global warming potential. Regarding social responsibility, green electrospinning creates green jobs from deployment of renewable technologies. It also supports a circular bioeconomy through valorization of waste resources. Furthermore, it upcycles agricultural residues that otherwise emit greenhouse gases. Overall, green electrospinning focuses on conservation of resources, clean manufacturing, renewable energy, carbon mitigation, and socioeconomic benefits to achieve sustainability across environmental, economic, and social dimensions.

## 5. Future Perspectives and Challenges

The discussion regarding the environmental footprint of green electrospun nanofiber materials underscores the stark contrast between conventional electrospinning practices and more sustainable approaches. Conventional methods heavily rely on toxic chemicals

and energy-intensive processes, which collectively contribute to a substantial carbon footprint and ecological harm. In contrast, green electrospinning techniques are engineered to mitigate these environmental impacts through a multifaceted approach. Firstly, green electrospinning prioritizes the use of renewable materials, drawing from biopolymers like cellulose, gelatin, and chitosan, which can be readily sourced from agricultural and marine waste. This not only ensures a consistent and sustainable supply of raw materials, but also reduces the burden on natural resources. Secondly, the elimination of hazardous chemicals from the production process is a key facet of green electrospinning. By eschewing organic solvents, it sidesteps toxicity concerns linked to their manufacture, use, and disposal, thus simplifying regulatory compliance for various applications. Thirdly, green electrospinning significantly curtails energy requirements. Innovative technologies such as solar, centrifugal, and near-field electrospinning substantially reduce electricity consumption when compared to the high-voltage demands of traditional methods, thereby lowering carbon emissions. Fourthly, certain green electrospinning processes synergize nanofiber fabrication with wastewater treatment or carbon dioxide sequestration, effectively converting waste into a valuable resource. This approach enhances overall eco-efficiency. Lastly, green electrospinning facilitates the development of biodegradable nanofibers, allowing them re-entrance to the biosphere after their intended use [62,63,65,66]. This closed-loop approach to material utilization exemplifies a commitment to enhancing material lifecycles. In summation, green electrospinning strategies collectively reduce reliance on nonrenewable resources, minimize pollution, tap into agricultural and industrial byproducts, and promote more circular material economies. This holistic approach effectively lessens the environmental impacts associated with nanomanufacturing, aligning with the imperative to tread lightly on our planet while advancing cutting-edge technology.

The sustainability aspects of green electrospinning technologies encompass a comprehensive range of considerations. These aspects, as identified through research, highlight the commitment of green electrospinning to environmental responsibility and social well-being [58–60]. Resource efficiency is a cornerstone of green electrospinning, achieved through the utilization of renewable biomass sources that do not encroach upon food production. Moreover, it capitalizes on recovering materials from waste streams, including valuable resources like lignocellulose and food processing residues. By promoting closed-loop manufacturing, green electrospinning fosters designs that facilitate disassembly and remanufacturing, minimizing waste and maximizing resource use. Cleaner production is another vital dimension, as green electrospinning eliminates toxic organic solvent emissions and wastewater effluents. It actively incorporates wastewater treatment processes and the removal of volatile compounds, while favoring aqueous, solvent-free, and mechanistic fabrication routes to ensure minimal environmental impact. Energy efficiency is optimized through the adoption of solar, centrifugal, and near-field methods, which significantly reduce electrical energy consumption. Furthermore, green electrospinning explores the integration of renewable energy sources such as solar and wind, aligning with sustainable energy generation applications to reduce reliance on nonrenewables. Environmental performance is a pivotal concern, with green electrospinning yielding fully biodegradable nanofibers that seamlessly re-enter the biosphere. It also contributes to carbon sequestration through the use of algal and bacterial cultures in production processes, actively mitigating greenhouse gas emissions. Comprehensive lifecycle impact assessments, particularly in terms of global warming potential, provide a holistic view of its environmental contributions [67,69,71]. Social responsibility is underscored through the creation of green jobs stemming from the deployment of renewable technologies in the green electrospinning sector. This approach not only supports a circular bioeconomy, but also upcycles agricultural residues that would otherwise emit greenhouse gases, enhancing its role in mitigating climate change. In sum, green electrospinning stands as a holistic approach, encapsulating resource conservation, clean manufacturing practices, the embrace of renewable energy, carbon mitigation, and socioeconomic benefits. These facets collectively contribute to sus-

tainability across environmental, economic, and social dimensions, marking a conscientious stride towards a more environmentally responsible and socially equitable future.

While significant strides have been made in the field of green electrospinning in recent years, there remain substantial opportunities for advancing this technology through targeted research endeavors. Several promising directions for future research have been identified [67,69,71]. Firstly, the development of biopolymers sourced from waste materials offers a promising avenue. Tailoring the structures and properties of these biopolymers could lead to the creation of next-generation sustainable materials. Additionally, the formulation of multifunctional composite materials would expand their applicability across various domains. Advanced process optimization is another critical area of focus. Techniques like aqueous electrospinning require comprehensive parametric studies, modeling, and scale-up experiments to fully unlock their potential. Integrating these manufacturing processes holds the potential to enhance efficiency and scalability. Further exploration of multifunctional materials engineering is essential, particularly in the development of stimuli-responsive fibers for applications like controlled theranostics and the integration of electronic components into composites. Green electrospinning can also make significant contributions to sustainable energy and environmental applications. Coupling nanofiber technologies with fields such as water treatment, desalination, and energy storage could yield impactful and environmentally friendly solutions. Conducting comprehensive life cycle assessments will provide meaningful sustainability metrics, enabling researchers to benchmark progress effectively. Collaboration with the industry for technology transfer and commercialization of green nanofibers will also support their widespread adoption. Continued focus on biopolymer design, process development, multifunctionality, renewable applications, environmental impacts, and commercialization pathways can push the boundaries of green electrospinning, aligning with the global movement towards sustainable nanomanufacturing. However, there are challenges that must be addressed to fully realize the potential of green electrospinning, as identified in the literature [73–75]. These challenges include precise control over biopolymer properties, sourcing sufficient biomass inputs, scaling up green techniques, maximizing sustainability metrics while maintaining functionality, and gaining industrial adoption while overcoming reliance on petrochemicals. New standards for evaluating biocompatibility and environmental profiles may also pose regulatory challenges. To overcome these challenges, researchers can focus on advancing biomaterials science to engineer optimal biopolymer formulations, collaborate across sectors to strengthen biomass supply chains, develop new reactor and process designs for scalability, employ modeling and machine learning for materials-by-design approaches, and foster partnerships to validate sustainability gains and drive technology transfer. Progress in these areas promises to fully realize the benefits of green electrospinning, expand sustainable nanomanufacturing, and establish it as a mature alternative beyond basic research, ushering in a forward-looking future.

## 6. Conclusions

In summary, this review provided a comprehensive overview of the pivotal field of green electrospinning in the context of sustainable nanomaterial production, yielding several key conclusions. Traditional electrospinning methods, reliant on toxic solvents and energy-intensive processes, have spurred the quest for greener alternatives. A range of techniques have emerged, optimizing parameters through aqueous formulations, solvent elimination, reduced voltages, and the integration of renewable energy sources. Biopolymers sourced from renewables, including cellulose, collagen, and gelatin, display desirable properties such as biodegradability and biomimicry when electrospun. Moreover, synthetic bioplastics and their composites hold promise as sustainable materials. Green manufacturing approaches have been instrumental in minimizing environmental impacts through strategies like solvent removal, the use of renewable feedstocks, waste utilization, and closed-loop material design, ultimately enhancing eco-efficiency compared to conventional techniques. Green electrospinning technologies have found applications in fields such as

biomedical nanofibers, controlled drug delivery, and environmental remediation, effectively addressing existing challenges. However, there is still a need for further work in materials engineering, scale-up processes, multifunctionality realization, and comprehensive life cycle sustainability assessments. In conclusion, the development of green electrospinning holds the potential to revolutionize sustainable nanomanufacturing on an industrial scale. Future directions should center on technology optimization and commercialization to fully unlock its beneficial impacts. This paradigm shift towards green electrospinning can contribute significantly to sustainable nanomaterial development and the broader circular bioeconomy, replacing nonrenewable inputs with abundant waste resources and driving the transition towards a low-carbon biomanufacturing future. By strategically designing products for recyclability and fostering cross-disciplinary collaboration, the field can advance and demonstrate its full life cycle sustainability benefits, ultimately offering solutions to pressing global challenges, including pollution, resource security, and climate change mitigation. Early commercial successes will signal the market viability of renewable nanotechnologies, further accelerating the transition towards a sustainable bioeconomy paradigm across industries. Continued progress in optimizing manufacturing processes and biomaterials while maintaining performance holds the promise of driving systemic changes towards a more sustainable future.

**Author Contributions:** Conceptualization, E.B. and O.D.; methodology, K.B.; software, W.M.N.W.N.; validation, I.E., M.A. and S.N.; formal analysis, M.D.; investigation, C.Y.; resources, N.A. All authors have read and agreed to the published version of the manuscript.

**Funding:** This research received no external funding.

**Institutional Review Board Statement:** Not applicable.

**Informed Consent Statement:** Not applicable.

**Data Availability Statement:** Not applicable.

**Conflicts of Interest:** The authors declare no conflict of interest.

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
