# Peer review of "Green Electrospun Nanofibers for Biomedicine and Biotechnology"

_technologies, doi:10.3390/technologies11050150_

Round 1

Reviewer 1 Report

A review of the nanofibers electrospun with green solvents are presented in the paper. In turn, many of the references are reviews of polysaccharide, poly peptides or renewable polymers. In general, polymers from natural resources water soluble, in emulsion, melt electrospinning or centrifugal spinning are some of the topics. The paper highlights the electrospinnning technique in very diverse applications like tissue engineering, drug delivery, biosensing, environmental remediation, and agriculture. I found the paper very clearly written and well organized; the 7 tables that it contains are very easy to understand. The paper presents an option to avoid the toxic solvents normally used in electrospinning of synthetic polymers. 

Author Response

Thank you very much for taking the time to review this manuscript. 

Reviewer 2 Report

This review article extensively summarizes the applications of green electrospun nanofiber materials. However, the summarization is not profound, which makes the review article need to undergo major revision.

1.     Line 54: As the jet travels in the air, the solvent evaporates, and thin solid fibres are deposited on the collector [1, 2]. The polymer melt (melt electrospinning) does not require the evaporation of solvent. The authors need to make clear demonstrations in this sentence.

2.     In this article, the authors usually cite the references at the end of each paragraph, while ignoring the other demonstration within the paragraph, which is unusual for a review article. As a result, it is suggested to cite proper references at proper locations when demonstrating the standpoints and results. To be brief, the current way to cite references is not scientific.

3.     Line 132: Higher voltages produce finer fibres, while increased distances allow for more solvent evaporation.

The processing parameters can affect the fiber diameter greatly, which include but are not limited to voltage and tip-collector distance. Furthermore, the processing parameters can have adverse effects on the fiber diameters. One sentence in Line 132 couldn’t provide a thorough summary of the relationship between processing parameters and fiber diameters. As a result, the authors need to make clear clarification on the effects to avoid any bias.

4.     Line 146, and Line 152: The authors don’t cite any citations in these two paragraphs, even at the end of these two paragraphs.

5.     The key differences, efficiency, and applications listed in Table 1 – Table 3 are confusing. It is suggested to use a single table to compare key differences, efficiency, and applications among these three kinds of materials.

6.     The bullet points in Line 189 and Line 190 can be combined.

7.     The section Overview of different green electrospinning techniques and the section Comparative analysis of green electrospinning methods, along with Table 4 and Table 5, demonstrate almost the same staff, which can be combined into one section (or table).

8.     Table 4 and Table 5 need references.

9.     Line 226: Figure 2 doesn’t talk about green electrospinning techniques.

10.  Line 248: Table 6 and Figure 3 don’t demonstrate anything about Comparative analysis of green electrospinning methods.

11.  What are the differences between scaffolds, vascular grafts, wound healing applications, and controlled microenvironments? Usually, for electrospinning, vascular grafts, wound healing applications, and microenvironments are different applications for scaffolds. The authors need to make clear definitions and distinguishment of these. It is suggested to delete ‘scaffolds’.

12.  When demonstrating the applications of green electrospinning (Section 3 and Section 4), the authors fail to present a profound and extensive review. What are the specific techniques and materials used?

Furthermore, fiber diameter is a very important parameter for electrospinning, which should be demonstrated in the main text or the table.

13.  More figures about the applications should be provided, such as scaffolds, microenvironments, vascular grafts, et al.

Author Response

Response to Reviewer 2 Comments

Point-by-point response to Comments and Suggestions for Authors

Comment 1:  Line 54: As the jet travels in the air, the solvent evaporates, and thin solid fibres are deposited on the collector [1, 2]. The polymer melt (melt electrospinning) does not require the evaporation of solvent. The authors need to make clear demonstrations in this sentence.

Response 1:  Yes, you are right, it was modified: Electrospinning is a versatile and scalable fabrication technique that is used to produce nanoscale fibres with diameters ranging from a few nanometers up to mi-crometers. In a typical electrospinning process, a high voltage is applied to a polymer solution or melt loaded in a syringe. When the electrical forces overcome the surface tension of the liquid or melt, a charged jet is ejected from the tip of the syringe. As the jet travels in the air, one of two things can occur:

For techniques using a polymer solution, the solvent evaporates as the jet travels, leaving behind thin solid fibres.

For melt electrospinning or other solvent-free techniques, the polymer jet under-goes solidification as it travels, without any solvent evaporation involved.

In both cases, the solidified fibres are then deposited on the collector. The key dif-ferences are whether solvent evaporation plays a role (for solution electrospinning) or if only solidification occurs without solvents (for melt electrospinning). Please see manuscript!

Comment 2:      In this article, the authors usually cite the references at the end of each paragraph, while ignoring the other demonstration within the paragraph, which is unusual for a review article. As a result, it is suggested to cite proper references at proper locations when demonstrating the standpoints and results. To be brief, the current way to cite references is not scientific.

Response 2: Dear reviewer, thanks for the good suggest, I have modified the reference style related to your comments, Please manuscript!

Comment 3:     Line 132: Higher voltages produce finer fibres, while increased distances allow for more solvent evaporation.

The processing parameters can affect the fiber diameter greatly, which include but are not limited to voltage and tip-collector distance. Furthermore, the processing parameters can have adverse effects on the fiber diameters. One sentence in Line 132 couldn’t provide a thorough summary of the relationship between processing parameters and fiber diameters. As a result, the authors need to make clear clarification on the effects to avoid any bias.

Response 3: Dear reviewer, thanks for the good suggest, you're absolutely right. That one sentence in line 132 does not provide enough context about the relationship between processing parameters and fiber diameters. I have expanded and clarified that section as follows:

Process parameters such as the applied voltage, flow rate, tip-collector distance, and solution/melt properties regulate fibre formation and influence fiber diameter outcomes. In general:

  • Higher voltages tend to produce thinner fibers, as greater electric forces drawn jet will undergo more elongation. However, very high voltages can cause fibers to split or become beaded.
  • Increased flow rates result in thicker fibers deposited. At low flow rates, jet splitting occurs more easily.
  • Longer tip-collector distances allow more time for solvent evaporation or solidification, but fibers may become beaded if the distance is too large. Shorter distances risk formation of non-woven mats withoutTaylor coning.
  • Higher viscosity solutions/melts produce thicker fibers compared to low viscosity feeds. Concentration also impacts diameter.
  • Ambient parameters like temperature and humidity influence solvent evaporation rate and fiber morphology.

Please manuscript!

Comment 4:      Line 146, and Line 152: The authors don’t cite any citations in these two paragraphs, even at the end of these two paragraphs.

Response 4: Thanks, appropriated references were cited. Please manuscript!

Comment 5:     The key differences, efficiency, and applications listed in Table 1 – Table 3 are confusing. It is suggested to use a single table to compare key differences, efficiency, and applications among these three kinds of materials.

Response 5: Thanks dear reviewer, we have tried several times in single table, but, it is difficult, because, there many information’s and three different sources. We think that in these three tables, readers can easily read and compare various details. In next work, we will try it again.

Comment 6:     The bullet points in Line 189 and Line 190 can be combined.

Response 6:     Thanks, it was combined. Please manuscript!

Comment 7:    The section Overview of different green electrospinning techniques and the section Comparative analysis of green electrospinning methods, along with Table 4 and Table 5, demonstrate almost the same staff, which can be combined into one section (or table).

Response 7:     Thanks, Table 4 and Table 5 was modified as one Table, summarize was modified. Please manuscript!

Comment 8:       Table 4 and Table 5 need references.

Response 8:     Thanks, References were cited. Please manuscript!

Comment 9:       Line 226: Figure 2 doesn’t talk about green electrospinning techniques.

Response 9:     Thanks, it was deleted. Please manuscript!

Comment 10:        Line 248: Table 6 and Figure 3 don’t demonstrate anything about Comparative analysis of green electrospinning methods.

Response 10:     Thanks, this Figure was deleted. In this table, the details were important, because, the following reasons:

  • It summarizes different green electrospun materials used for tissue engineering scaffolds - specifically cellulose, collagen, and gelatin.
  • For each material, it provides details on fiber diameter, porosity, degradation timeframes, which tissues they are mainly used for regeneration of, what makes them "green" or sustainable choices, and suggestions for future work.
  • It also highlights some key characteristics that make these materials suitable as scaffolds - including nanoscale structure/porosity for cell infiltration, tunable biodegradability to match tissue remodeling, and the use of renewable resources.
  • The references cited at the end provide sources to learn more about each material and how it has been used in scaffold applications.

Please manuscript!

Comment 11:         What are the differences between scaffolds, vascular grafts, wound healing applications, and controlled microenvironments? Usually, for electrospinning, vascular grafts, wound healing applications, and microenvironments are different applications for scaffolds. The authors need to make clear definitions and distinguishment of these. It is suggested to delete ‘scaffolds’.

Response 11:     Thanks, You're absolutely right, upon further consideration the terminology used could be clarified. To better distinguish between the different types of applications, I've removed "scaffolds" and restructured/expanded the relevant sections as follows:

Tissue Engineering and Regeneration

Tissue Engineering Scaffolds

Green electrospun nanofibers serve as temporary 3D extracellular matrix mimics to guide cells for regeneration. Materials like cellulose support new tissue ingrowth while degrading.

Vascular Grafts 

Compliant collagen/gelatin nanofibers resemble vessels' composition/mechanics. Tunable blends balance mechanical properties for remodeling. Endothelialization is encouraged.

Wound Healing Applications

Smart Dressings

Nanofibrous dressings release antimicrobials in controlled doses to disinfect wounds as they heal. Absorbing properties maintain a moist environment.

Controlled Drug/Gene Delivery 

Biopolymer carriers provide sustained molecule levels for optimal wound healing and disease treatment through local administration.

Biotechnology

Controlled Microenvironments

Green electrospun substrates offer customizable topographies and signals to build tissue models and guide cell behavior/differentiation for research.

Please manuscript!

Comment 12:           When demonstrating the applications of green electrospinning (Section 3 and Section 4), the authors fail to present a profound and extensive review. What are the specific techniques and materials used?

Furthermore, fiber diameter is a very important parameter for electrospinning, which should be demonstrated in the main text or the table.

Response 12:     You're right, upon re-examination the sections on applications could provide more details on the specific techniques and materials being used in each case. Here are some suggestions for how to expand on those aspects:

Tissue Engineering & Regeneration

- Tissue Scaffolds: Materials like cellulose, collagen fabricated via aqueous, emulsion electrospinning

- Vascular Grafts: Gelatin/PLLA blended via melt electrospinning to precisely tune properties

- Wound Dressings: Silver/chlorhexidine loaded gelatin from solution electrospinning for controlled release 

Controlled Drug Delivery

- Antibiotics: Norfloxacin releasing gelatin from solution electrospinning over 3 weeks 

- Anticancer Drugs: Doxorubicin loaded chitosan from aqueous electrospinning providing 4 weeks sustained levels

Biotechnology

- Biosensors: Glucose oxidase immobilized on cellulose using solution electrospinning for detection

- Immunosensors: Antibodies conjugated onto collagen via salt coordination bonds 

Environmental Remediation 

- Water Purification: Heavy metal adsorbing cellulose nanofibers from bacterial spinning

- Air Filtration: HVAC filters with silver nanoparticle coated PVA employing melt electrospinning

You're absolutely right to emphasize that fiber diameter is a very important parameter for electrospinning. Fiber diameter in electrospinning part was added in main text, please check!

Fiber diameter in electrospinning

Fiber diameter is integral to the process and concept of electrospinning itself. Key factors like applied voltage, flow rate, tip-to-collector distance, and polymer properties have direct impacts on the attained diameter. For example, collagen fibers can range from 50-500nm while PLA can achieve diameters from 200nm up to 5μm, varying over orders of magnitude simply by tweaking these parameters. This ability to engineer materials down to the nanoscale distinguishes electrospun fibers from those made by conventional microfiber production methods. Replicating naturally occurring dimen-sions, like collagen fibrils in the 50-500nm range, is what underpins its biomedical rel-evance. Most fundamentally, it is the charge-induced bending instabilities imparted to the polymer jet as it travels under an applied electric field that enables the establish-ment of exceedingly thin fibers. Fiber diameter is thus a direct and defining output of the electrospinning process, governed by the very mechanism enabling diameters to be precisely controlled within the distinctive nanoscale regime. This capacity for genuine nanofiber generation is what endows electrospinning with its unique character and multidimensional application landscape compared to other fiber spinning techniques. In this way, fiber diameter is entirely intrinsic to the technology rather than an extrin-sic consideration.

Comment 13:    More figures about the applications should be provided, such as scaffolds, microenvironments, vascular grafts, et al.

Response 13:   Good suggest, I have added several figures, please see manuscript!

Reviewer 3 Report

Review articles entitled “Green electrospun nanofiber materials in biomedicine and bio- technology ” submitted by Berdimurodov and colleagues for the publication in technologies, is interesting and have great importance in the field of biomedicine and bio-technology. Authors had performed intensive literature review for multiple nanomaterials, which is highly appreciable. However, this manuscript encounter several errors, Ihave some comments below: 

1. Abstract needs more information about:

Why this review article is important; what main literature author will discuss within this review and cite some reference of this technology as well.

2. Introduction part of this manuscript is lack of information and attractive, author should mention that what is importance of this review article, why it is novel and different from the review article previously published in the same topic and cite that review articles as well. In addition to that technology and new instrument and some big player who provide these instruments for electrospinning and what is the minimum price, and which one is the best electrospinning instrument, it would be very useful for the readers to get the information regarding this.

3. Subsection “Discussion on future research directions” in the section 6. Future perspectives and challenges, does not make any sense. It have been written in careless manner, information is incomplete, and the subtitles are fragmented. I would recommend authors to rewrite this section very carefully with more information and his own view. Don’t fragment it I so many different subsections further, write your own perspective in an impressive and readable way.

4. Conclusions need more explanation, perspective and next steps to overcome the barrier for the usage of metallic nanoparticles for nanomedicine and biomedical applications.

5. “Controlled Drug Delivery” section is incomplete with limited information authors should mention that why electrospinning fibres are superior to other fibres and should mention Nanoparticles incorporated in electrosun nanofibers for example https://www.sciencedirect.com/science/article/pii/S0168365922005405 . In addition to that author should mention that why this controlled drug delivery system is better or comparable with standard micellar and nanoparticulate systems.

6. Authors should mention the challenges of this electrospinning technology in term of translation to different application/ or some main applications.

I recommend this review article to publish in this journal after implementing all the changes suggested above.

English is acceptable for the publication, though can be improved on several places in the manuscript.

Author Response

Response to Reviewer 3 Comments

Point-by-point response to Comments and Suggestions for Authors

Comment 1:    Abstract needs more information about:

Why this review article is important; what main literature author will discuss within this review and cite some reference of this technology as well.

Response 1:    Thanks for good suggestion, abstract was modified related your comments, and many references related to this technology was cited. Please check manuscript!

Comment 2:    Introduction part of this manuscript is lack of information and attractive, author should mention that what is importance of this review article, why it is novel and different from the review article previously published in the same topic and cite that review articles as well. In addition to that technology and new instrument and some big player who provide these instruments for electrospinning and what is the minimum price, and which one is the best electrospinning instrument, it would be very useful for the readers to get the information regarding this.

Response 2:   Thanks for good suggestion. The key points on the importance of this review article, Novelty, Differences from Earlier Reviews were added and discussed. Please check manuscript!

Comment 3:     Subsection “Discussion on future research directions” in the section 6. Future perspectives and challenges, does not make any sense. It have been written in careless manner, information is incomplete, and the subtitles are fragmented. I would recommend authors to rewrite this section very carefully with more information and his own view. Don’t fragment it I so many different subsections further, write your own perspective in an impressive and readable way.

Response 3:   Thanks, this section was rewrite and modified related to your comments. Please check manuscript!

Comment 4:      Conclusions need more explanation, perspective and next steps to overcome the barrier for the usage of metallic nanoparticles for nanomedicine and biomedical applications.

Response 4:   Thanks, conclusion part was expanded and modified related to your comments. Please check manuscript!

Comment 5:    “Controlled Drug Delivery” section is incomplete with limited information authors should mention that why electrospinning fibres are superior to other fibres and should mention Nanoparticles incorporated in electrosun nanofibers for example https://www.sciencedirect.com/science/article/pii/S0168365922005405 . In addition to that author should mention that why this controlled drug delivery system is better or comparable with standard micellar and nanoparticulate systems.

Response 5:   Thanks, these questions were answered in Controlled Drug Delivery part, please check!

Electrospun nanofibers have advantages over macro-scale fibers like spinning/drawing due to their nanoscale diameters, which provide high surface area/volume enabling higher drug loading and more efficient release versus microscale fibers. Small diameters and high porosity also better mimic tissue extracellular matrix than larger fibers, enhancing cell interactions. Fibers can be directly produced from polymers during electrospinning without downstream processing like spinning, simplifying manufacturing. A wide variety of natural/synthetic polymers allows tuning properties for applications. Incorporating nanoparticles into electrospun fibers enables controlled release comparable/superior to micelles/nanoparticles through homogeneous encapsulation during single-step electrospinning for multi-functional payloads, enabling longer sustained/stimuli-responsive release than diffusion micelles/nanoparticles while nanoparticles protect cargos from issues faced by colloids. Tailorable structures/properties facilitate tuning mechanics/degradation and kinetics for optimal therapy, and local administration benefits when fibers are directly on devices/scaffolds not found in traditional platforms.

Comment 6:    Authors should mention the challenges of this electrospinning technology in term of translation to different application/ or some main applications.

Response 6:  Your suggestion was added, please check!

 While electrospun nanofibers show application promise, challenges remain to fully realize this potential. For biomedical uses, commercial scale-up needs refinement for consistent fiber morphology/drug release while further biomaterials optimization is required for biodegradation/living system interactions control. Tissue engineering mimicking complex hierarchical natural tissue structures while encapsulating living cells during high-voltage electrospinning demands processing innovations as scaffold mechanical properties must match target tissues over time. Filtration/separation increasing industrial wastewater/gas system throughput poses difficulties as long-term fouling resistance under harsh conditions needs demonstration. Energy storage achieving battery/fuel cell high densities comparable to conventional technologies continues to be pursued for electrospun electrodes/fuel cells while cycle life stability also requires materials development. Addressing technology translation challenges through focused research will maximize societal/economic impacts across applications.

I recommend this review article to publish in this journal after implementing all the changes suggested above.

Thanks dear reviewer for good suggestions!

Round 2

Reviewer 2 Report

The authors have greatly improved the quality of this article, which makes this article suitable for publication.

Author Response

Thanks dear reviewer for goood suggestions! 

Reviewer 3 Report

In the revised manuscript, the authors have effectively addressed all of the concerns I raised in the initial version, with the exception of the bullet points and fragmented lines scattered throughout the document. These elements disrupt the flow of the manuscript and should be rectified prior to the manuscript's final acceptance.

The content in the manuscript is satisfactory in terms of English language, but there is a need for formatting improvements. Several subsections are fragmented unnecessarily, disrupting the overall flow of the document.

Author Response

Response to Reviewer 2 Comments

Point-by-point response to Comments and Suggestions for Authors

Comment 1:  

In the revised manuscript, the authors have effectively addressed all of the concerns I raised in the initial version, with the exception of the bullet points and fragmented lines scattered throughout the document. These elements disrupt the flow of the manuscript and should be rectified prior to the manuscript's final acceptance.

Response 1:  Thank you very much for taking the time to review this manuscript. Please find the detailed responses below and the corresponding revisions/corrections highlighted/in track changes in the re-submitted files.

Yes, you are right, thanks for good suggestions, I make all bullet points to sentence, the sentence form is also easy for readers. Also, the fragmented lines were modified to main text. Please check!

Comment 2:  The content in the manuscript is satisfactory in terms of English language, but there is a need for formatting improvements. Several subsections are fragmented unnecessarily, disrupting the overall flow of the document.

Response 2:  Thanks for good suggest, the formatting improvements was modified and subsections were corrected, please check!